# Omega-3 Fatty Acids-Enriched Fish Oil Activates AMPK/PGC-1α Signaling and Prevents Obesity-Related Skeletal Muscle Wasting

**DOI:** 10.3390/md17060380

**Published:** 2019-06-25

**Authors:** Shing-Hwa Liu, Chen-Yuan Chiu, Lou-Pin Wang, Meng-Tsan Chiang

**Affiliations:** 1Graduate Institute of Toxicology, College of Medicine, National Taiwan University, Taipei 100, Taiwan; shinghwaliu@ntu.edu.tw; 2Department of Pediatrics, College of Medicine and Hospital, National Taiwan University, Taipei 100, Taiwan; 3Department of Medical Research, China Medical University Hospital, China Medical University, Taichung 404, Taiwan; 4Institute of Food Safety and Health, College of Public Health, National Taiwan University, Taipei 100, Taiwan; chenyuanchiu@ntu.edu.tw; 5Department of Food Science, College of Life Science, National Taiwan Ocean University, Keelung 202, Taiwan; gmov2504@gmail.com

**Keywords:** fish oil, obesity, skeletal muscle wasting, AMP-activated protein kinase

## Abstract

Obesity is known to cause skeletal muscle wasting. This study investigated the effect and the possible mechanism of fish oil on skeletal muscle wasting in an obese rat model. High-fat (HF) diets were applied to induce the defects of lipid metabolism in male Sprague-Dawley rats with or without substitution of omega-3 fatty acids-enriched fish oil (FO, 5%) for eight weeks. Diets supplemented with 5% FO showed a significant decrease in the final body weight compared to HF diet-fed rats. The decreased soleus muscle weights in HF diet-fed rats could be improved by FO substitution. The decreased myosin heavy chain (a muscle thick filament protein) and increased FOXO3A and Atrogin-1 (muscle atrophy-related proteins) protein expressions in soleus muscles of HF diet-fed rats could also be reversed by FO substitution. FO substitution could also significantly activate adenosine monophosphate (AMP)-activated protein kinase (AMPK) phosphorylation, peroxisome-proliferator-activated receptor-γ (PPARγ) coactivator 1α (PGC-1α), and PPARγ protein expression and lipoprotein lipase (LPL) mRNA expression in soleus muscles of HF diet-fed rats. These results suggest that substitution of FO exerts a beneficial improvement in the imbalance of lipid and muscle metabolisms in obesity. AMPK/PGC-1α signaling may play an important role in FO-prevented obesity-induced muscle wasting.

## 1. Introduction

Skeletal muscle is one of the main organs of glucose and lipid metabolism in the body. Various pathological conditions, including obesity, diabetes, cancer, and ageing, display a characteristic feature of skeletal muscle mass loss [1]. The influence of dietary westernization and decreased physical activity causes the body composition to gradually change so that body weight and fat accumulation in the skeletal muscles and abdominal visceral layer and organs are gradually increased [2]. Studies have shown that high-fat (HF) diet intake reduces the content and function of mitochondria in skeletal muscles [3,4]. HF diet feeding can induce skeletal muscle atrophy in animal models [5,6]. Palmitate has been found to trigger the pro-atrophic genes (e.g., muscle atrophy f-box (MAFbx)/Atrogin-1) expression, which are concomitant with increased nuclear localization of Forkhead O (FOXO) 3 in skeletal muscle cells [7]. Conversely, substitution of omega-3 polyunsaturated fatty acids (PUFAs) protected against the palmitate-induced atrophy in C2C12 myotubes [8].

Fish oil is known to be enriched omega-3 PUFAs, such as docosahexaenoic acid (DHA) and eicosapentaenoic acid (EPA) [9]. Dietary substitution of omega-3 PUFAs has been suggested to maintain the balance of energy metabolism in muscles that can reduce the accumulation of fatty acids, improve the function of mitochondria, and provide the integrity of phospholipid structure on muscle cell membrane [4,10,11]. It has also been reported that the uptake of omega-3 PUFAs can improve insulin resistance, especially in liver and skeletal muscle, in HF diet-fed rats [12]. Fish oil intake has also been found to increase insulin sensitivity of skeletal muscle and promote lipid oxidation and further reduce fatty acid accumulation [11]. Fish oil-derived omega-3 PUFAs intake has been shown to slow age-associated muscle mass and function loss in healthy older adults [13]. Dietary substitution with fish oil-derived omega-3 PUFAs has been suggested to possess the potential for the prevention and treatment of sarcopenia in older people [14]. However, the effects and regulatory signals of fish oil on skeletal muscle wasting are still unclear. In the present study, we, therefore, investigated the beneficial effect and the possible regulatory signals of fish oil on skeletal muscle wasting in an obese rat model.

## 2. Results and Discussion

Obesity and type 2 diabetes as metabolic disorders have become a public health concern due to aging, physical inactivity, and dietary consumption of high-energy-density foods in Westernized countries and developed countries worldwide, leading a raised risk of mortality and morbidity. The deteriorated manifestation of these metabolic disorders with the dietary supplementation of HF diets is targeted not only on well-known cardiovascular organs but also on the skeletal muscles. Several studies have confirmed that HF diets induce skeletal muscle wasting and dysfunction in animals and in clinic. Shortreed et al. have indicated the continued exposure to HF diets would contribute to a significant decrease in myofiber contractile force due to the impairment of oxidative capacity [15], which is consistent with the observation in obese older adults suggested by Choi et al. (2015) [16]. In addition, Eshima et al. discovered morphological changes of myofiber composition of extensor digitorum longus muscle in HF diet-fed mice for 12 weeks, leading to the impairment of muscle contractile force [17]. Moreover, the mechanisms of skeletal muscle wasting and dysfunction in HF diet-induced obesity are positively correlated among several risk factors and signaling pathways including cardiometabolic risk factors [18], elevating inflammatory factors (such as tumor necrosis factor-α (TNF-α) levels), the activation of oxidative stress [5], autophagy [19,20], and ubiquitin–proteasome pathway [5], the down-regulation of AMPK/PGC-1 [21,22] and IGF/PI3K/Akt/mTOR signaling pathway [21,23], leading to skeletal muscle atrophy and apoptosis.

Fish oil with an abundance of omega-3 PUFAs has been known as a beneficial dietary supplement for the improvement of metabolic disorders [11,12]. In our previous study, we found that fish oil substitution effectively displayed hypolipidemic effect through the alleviation of HF diet-induced imbalance of lipid metabolism, which involved the coregulation of hepatic and intestinal microsomal triglyceride transfer protein (MTTP) and other lipid-transport-related signals with the activation of AMPKα/PPARα signaling [24]. In this study, we further investigated the effect and mechanism of fish oil on skeletal muscle metabolism and muscle wasting in HF diet-fed rats. Diets supplemented with 5% fish oil showed a significant decrease in the progression of body weight (Figure 1A-a) and final body weight (Figure 1A-b) compared to HF diet-fed rats. Since the present study is a continuation of our previous publication [24], the progression of the body weights of animals (Figure 1A-a) is the same as in our previous study [24]. Moreover, the decreased soleus muscle weights (Figure 1B) and muscle fiber atrophy (decreased cross-sectional area) (Figure 2) were shown in HF diet-fed rats, which could be effectively improved by fish oil substitution. These results indicate that fish oil possesses the potential to alleviate the skeletal muscle wasting in HF diet-induced obese condition.

Our results are consistent with a previous study carried out by Martins et al. [25]. As it is known that FO exhibits an abundance of omega-3 PUFAs, including eicosapentaenoic acid (EPA) and docosahexaenoic acid, several studies have indicated EPA and DHA both exert protective effects against HF diet-induced muscle deterioration, including the loss of myogenic capacity and degradation through AMPK activation [26,27,28,29,30], and our findings are similar to these studies that omega 3-rich FO exerts beneficial effects and similar mechanisms on muscle wasting in HF diet-induced obese rats. Moreover, recently, Lee et al. and Oh et al. have provided the evidence that conjugated linoleic acids and omega-3 PUFAs could attenuate muscle protein degradation and improve muscle strength in HF diet-induced obese mice [31,32]. Therefore, we will investigate the similar mechanism of the beneficial effects under the supplementation with conjugated linoleic acids and omega-3 PUFAs in HF diet-induced obese animals as in the present study, in the future.

Ubiquitin-proteasome-dependent proteolysis plays an important role in muscle wasting/atrophy. The muscle-specific ubiquitin ligases (e.g., MAFbx/Atrogin-1) are known to contribute to muscle protein degradation [33]. The activation of FOXO3 signaling can act as Atrogin-1 transcription and induce atrophy of myotubes and muscle fibers [34]. We next tested the involvement of these signaling molecules in muscles of HF diet-fed rats with or without fish oil substitution. As shown in Figure 3, the decreased MHC (a muscle thick filament protein) protein expression and increased FOXO3A and MAFbx/Atrogin-1 (muscle atrophy-related proteins) protein expression was observed in the soleus muscles of HF diet-fed rats, which could be significantly reversed by fish oil substitution. Moreover, the activation of AMPK in skeletal muscle is known to enhance glucose uptake, fatty acid oxidation, and mitochondrial biogenesis [35]. Moreover, Atherton et al. (2005) indicate that the inhibition of AMPK/mTOR signaling pathway (i.e., the inhibition of phosphorylated AMPK) would lead to the inhibition of protein synthesis and lack of hypertrophic muscles [36]. PGC-1α is known to be important downstream signaling of AMPK in skeletal muscles [37,38]. PGC-1α has been found to protect skeletal muscle against atrophy via inhibiting FOXO3/atrophy-specific gene transcription signaling [39]. We, therefore, investigated the role of AMPK/PGC-1α in muscle wasting of HF diet-fed rats with or without fish oil substitution. As shown in Figure 4, fish oil substitution significantly activated AMPK phosphorylation and increased PGC-1α and PPARγ protein expressions in soleus muscles of HF diet-fed rats. These results indicate that fish oil prevents against muscle wasting in HF diet-induced obese condition via activating AMPK/PGC-1α signaling.

In addition to FO as a positive substitution in the enforcement of skeletal muscle mass and strength, creatine monohydrate has been used as a nutritional supplement to preserve and promote skeletal muscle mass and strength for either young adults or aged adults [40,41]. The hypertrophic effect of creatine has been investigated and it was found that creatine could enhance protein synthesis under the regulation between IGF/Akt/mTOR signaling pathway and myogenic regulatory factors including MyoD and MHC in cultured cells [42,43] and human subjects [44,45]. However, in the present study, FO substitution could respectively reverse HF-inhibited MHC protein expression and HF-induced Atrogin-1 protein expression through the activation of AMPKα/PGC-1 signaling pathway, suggesting that FO enhances the hypertrophic effect against HF-induced muscle wasting under the activation of nutrient- and energy-sensing protein, AMPK, in skeletal muscle. Nevertheless, the involvement of IGF/Akt/mTOR signaling pathway in the hypertrophic effect of FO could not be excluded, and we will further investigate the role of FO in IGF/Akt/mTOR signaling pathway against HF-induced muscle wasting in the future.

According to previous studies, high-fat diets or high-carbohydrates contributing to the development of obesity may suppress fat oxidation through down-regulation of LPL expression in skeletal muscle [46,47]. Moreover, Boivin et al. have suggested that insulin resistance induced by HF diets and high-carbohydrate diets in rats may decrease skeletal muscle LPL activity [48]. Crunkhorn et al. have found that PPARγ and PGC-1α expressions were significantly decreased in the skeletal muscles of HF diet-fed mice [3]. AMPK-activated PPARγ1 signaling has been demonstrated to be involved in exercise-increased LPL gene expression in skeletal muscle [49]. Muscle LPL can hydrolyze triglycerides, which are contained in chylomicrons and VLDL, to yield fatty acids [50]. The overexpression of muscle-specific LPL has been shown to prevent HF diet-induced obesity [46]. In the present study, we found that dietary fish oil substitution significantly increased the LPL mRNA expression in soleus muscles of HF diet-fed rats (Figure 5).

## 3. Materials and Methods

### 3.1. Animals

Male Sprague–Dawley rats (six-week-old) were obtained from BioLASCO Taiwan Co., Ltd. (Taipei, Taiwan). Rats fed with a chow diet (Rodent Laboratory Chow, Ralston Purina, St. Louis, MO) were acclimatized for one week. Rats were randomly divided into three groups (*n* = 6 of each group): standard rodent diet (NC), HF diet (HF), and HF diet with 5% fish oil (Sentosa Co., Taipei, Taiwan) (HF+FO). The formulation of the diets and the fatty acid composition of fish oil were displayed as described in our previous study [24]. Briefly, the diets contained lard 3% + soybean oil 2% and others for the NC group, lard 15% + soybean oil 2% and others for the HF group, and lard 10% + soybean oil 2% + 5% fish oil and others for the HF + FO group. Fish oil contained C20:5 (EPA) 30.6% and C22:6 (DHA) 19.2% and others. Rats were kept in individual stainless-steel cages with controlled temperature (23 ± 1 °C), humidity (40–60% relative humidity) and lighting (12 h light/dark cycle). Following 12-h fasting, rats were sacrificed under anesthesia after eight weeks of experimental administration. The soleus muscles [51,52] were isolated, weighed, flash-frozen, and stored at −80 °C until analysis. The Animal House Management Committee of the National Taiwan Ocean University approved this study (permission number 105016). The experimental protocol of animal treatment was in accordance with the guidelines for the care and use of laboratory animals [53].

### 3.2. Histological Examination

The soleus muscles were fixed in formalin (10%) and then embedded in paraffin. The soleus muscle sections with 5-μm thick were stained with hematoxylin and eosin (H&E). A Nikon Eclipse TS100 microscope equipped with a Nikon D5100 digital camera was used to observe and photograph the H&E-stained sections. The cross-section areas (CSA) of individual myofibres were counted and calculated using the image J 1.48 software (National Institutes of Health, Bethesda, MD, USA) in five random fields of each section, as described previously [23,54].

### 3.3. Western Blot Analysis

Western blotting was determined as described previously by Chiu et al. [24]. Briefly, tissue lysates were collected by using the homogenized buffer (5 mM HEPES, 320 mM sucrose, 150 mM NaCl, 13 mM Tris-HCl, 13 mM EDTA) with a commercial cocktail of protease and phosphatase inhibitors (78443; 1:100 dilution) (Thermo Fisher Scientific, Waltham, MA, USA). After centrifugation, the supernatants were harvested and the protein concentration was determined by the BCA protein assay kit (23225; Thermo Fisher Scientific). Tissue proteins (50–100 μg) and prestained protein ladders (10 to 180 kDa or 250 kDa) (26616 or 26619; Thermo Fisher Scientific) were loaded in SDS-PAGE gel (8%–12%) and then transferred onto polyvinylidene difluoride membranes (Bio-Rad, Hercules, CA, USA). The membranes were blocked with 5% non-fat dry milk (Fonterra Brands, Taipei, Taiwan) or 3% BSA (Sigma-Aldrich, St. Louis, MO, USA) in 0.2% TBS-T buffer for at least 1 h, and then probed with primary antibodies for phosphorylated adenosine monophosphate (AMP)-activated protein kinase α (p-AMPKα) (Thr172) (2535; 1:1000 dilution) (Cell Signaling Technology, Danvers, MA, USA), AMPKα (5831S; 1:1000 dilution) (Cell Signaling Technology), peroxisome-proliferator activated receptor-γ (PPARγ) coactivator 1α (PGC-1α) (ab54481; 1:1000 dilution) (Abcam, Cambridge, MA, USA), PPARγ (sc-7273; 1:1000 dilution) (Santa Cruz Biotechnology, Santa Cruz, CA, USA), myosin heavy chain (MHC) (A4.1025 antibody, anti-all MHC, Merck Millipore, Burlington, Massachusetts, USA), Forkhead O (FOXO) 3A (ab53287; 1:1000 dilution), phosphorylated FOXO3A (Ser253) (ab154786; 1:1000 dilution) (Abcam), MAFbx/Atrogin-1 (ab74023; 1:1000 dilution) (Abcam), and β-actin (sc-47778; 1:1000 dilution) (Santa Cruz Biotechnology) at 4℃ overnight. Anti-rabbit (7074S; 1:5000 dilution) (Cell Signaling Technology) and anti-mouse (7076S; 1:5000 dilution) (Cell Signaling Technology) horseradish peroxidase-conjugated secondary antibodies (anti)were then used to probe the membranes. The bindings were determined by an enhanced chemiluminescence kit (BioRad Laboratories, Redmond, WA, USA) and then exposed to X-ray film (Fujifilm, Tokyo, Japan). The densitometric quantification of bands was determined by Image J 1.51 software (National Institutes of Health, Bethesda, MD, USA).

### 3.4. Quantitative Reverse Transcription Polymerase Chain Reaction (qRT-PCR) Analysis

The qRT-PCR was performed as described previously by Chiu et al. [24]. Briefly, a TRIzol kit (Life Technologies, Carlsbad, CA, USA) and an ABI StepOne™ Real-Time PCR system and StepOne 2.1 software (Applied Biosystems, Foster City, CA, USA) were used to perform total RNA extraction and qRT-PCR analysis. The 0.5–1 μg total RNA extracted from skeletal muscle tissues was converted into complementary DNA (cDNA) by avian myeloblastosis virus reverse transcriptase, and then the amplification of cDNA was determined by real-time SYBR Green PCR reagent (Life Technologies, Carlsbad, CA, USA) with specific primers: GAPDH (as a reference gene) (forward: CTGGAGAAACCTGCCAAGTATGAT; reverse: TTCTTACTCCTTGGAGGCCAGTA), lipoprotein lipase (LPL) (forward: GCAGGAAGTCTGACCAACAAG; reverse: CTTCACCAGCTGGTCCACAT). The relative quantification of gene expression was determined by the comparative threshold cycle method (_ΔΔ_CT) in which the expression level of each target gene was normalized to the levels of GAPDH. 2^−ΔΔCT^ was used to express as fold changes of mRNA expression levels relative to the HF group.

### 3.5. Statistical Evaluation

Data are presented as the mean ± standard deviation (SD). The one-way analysis of variance (ANOVA) followed by Turkey’s test for multiple comparisons using GraphPad Prism V6.0 software with a significance threshold of 0.05.

## 4. Conclusions

The results of the current study suggest that dietary substitution of fish oil exerts a beneficial improvement in the muscle wasting in obesity in an animal model. We further demonstrate that AMPK/PGC-1α-suppressed FOXO3/atrophy-specific gene transcription signaling plays an important role in fish oil-prevented obesity-induced muscle wasting. Moreover, fish oil substitution-increased PPARγ expression in response to AMPK activation induces LPL gene expression, which is associated with metabolic regulation in skeletal muscle. However, several limitations exist in the current study, such as other atrophy-related proteins MuRF-1 (another muscle-specific ubiquitin ligase) and autophagy-related proteins LC3b and p62 and down-stream genes or proteins of AMPKα not being included. Those may weaken the evidence of the conclusion.

## Figures and Tables

**Figure 1 marinedrugs-17-00380-f001:**
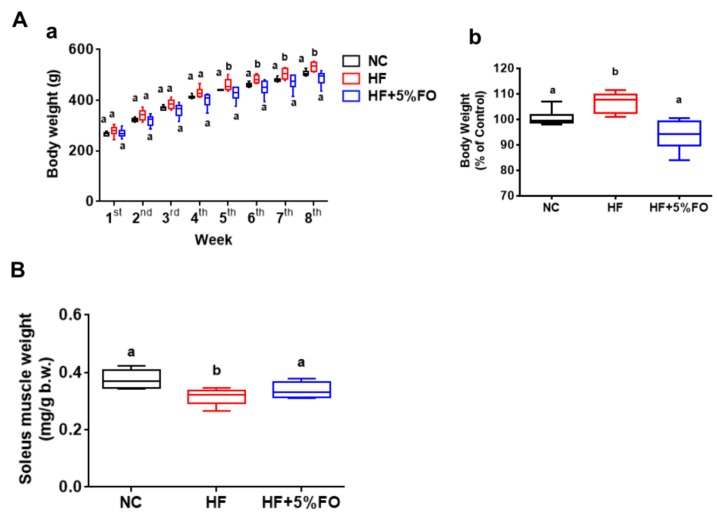
Effects of fish oil on body weight and muscle weight in high-fat diet-fed rats. The progression of body weight (**A**-a) [24] and the changes of the final body weights (**A**-b) and soleus muscle weights (**B**) in rats fed with standard diet (NC), high-fat diet (HF) in the presence or absence of 5% fish oil (FO) for eight weeks were shown. Results are expressed as mean ± S.D. for each group (*n* = 6). Values with different letters were significantly different, *P* < 0.05 (ANOVA with post-hoc Tukey’s test).

**Figure 2 marinedrugs-17-00380-f002:**
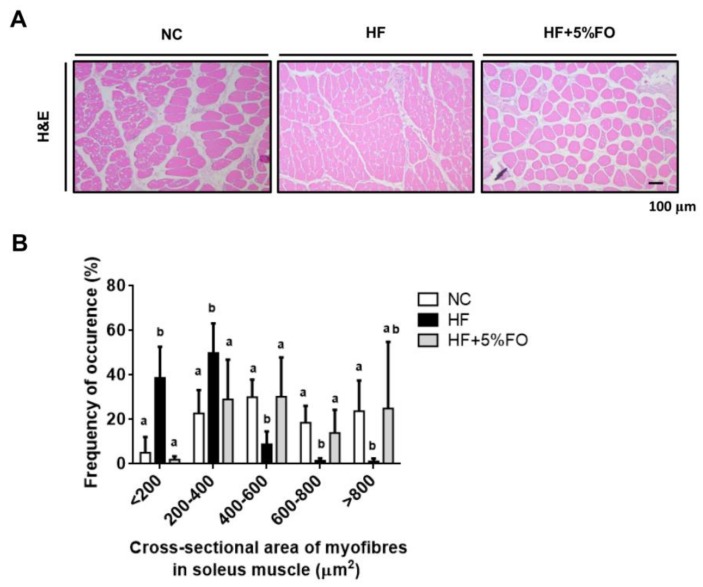
Effects of fish oil on morphology of high-fat diet-fed rats. Histological analysis of soleus muscle dissected from rats fed with different experimental diets was determined by hematoxylin and eosin (H&E) staining. Representative H&E stained images of soleus muscles were shown (**A**). The cross-sectional area of myobibres in soleus muscles was calculated (**B**). Results are expressed as mean ± S.D. for each group (*n* = 6). Values with different letters were significantly different, *P* < 0.05 (ANOVA with post-hoc Tukey’s test).

**Figure 3 marinedrugs-17-00380-f003:**
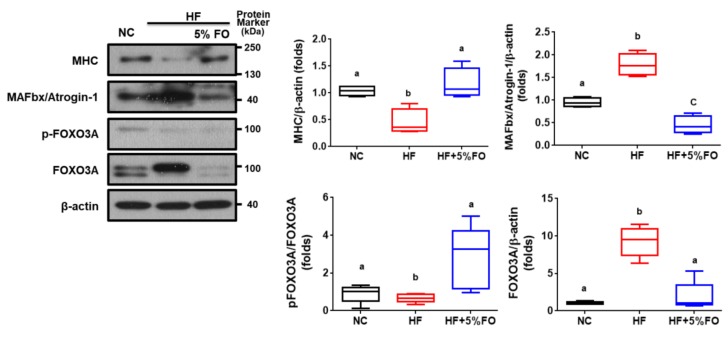
Effects of fish oil on the expressions of muscle thick filament protein and atrophy-related proteins in soleus muscles of high-fat diet-fed rats. Rats fed with different experimental diets (standard diet (NC), high-fat diet (HF) in the presence or absence of 5% fish oil (FO) for eight weeks. Protein expressions of myosin heavy chain (MHC), MAFbx/Atrogin-1, Forkhead O (FOXO) 3A, and phosphorylated FOXO3A in the soleus muscles were determined by Western blotting. Densitometric analysis for protein levels corrected to each internal control was shown. Results are expressed as mean ± S.D. for each group (*n* = 6). Values with different letters were significantly different, *P* < 0.05 (ANOVA with post-hoc Tukey’s test).

**Figure 4 marinedrugs-17-00380-f004:**
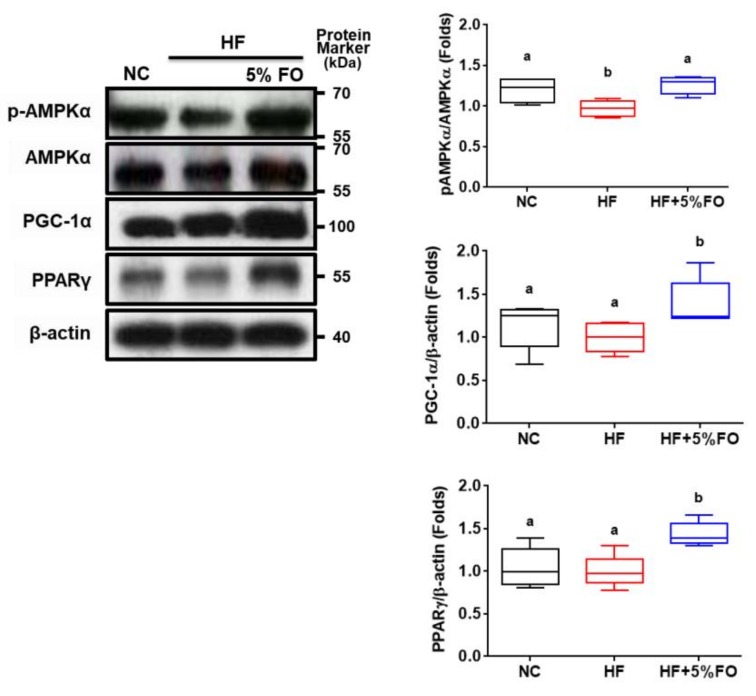
Effects of fish oil on the protein expressions of signaling molecules in muscles of high-fat diet-fed rats. Rats fed with different experimental diets (standard diet (NC), high-fat diet (HF) in the presence or absence of 5% fish oil (FO)) for eight weeks. Protein expressions of phosphorylated AMPKα/AMPKα, PGC-1α, and PPARγ in the soleus muscles were determined by Western blotting. Densitometric analysis for protein levels corrected to each internal control was shown. All data are expressed as mean ± S.D. for each group (*n* = 6). Values with different letters were significantly different, *P* < 0.05 (ANOVA with post-hoc Tukey’s test).

**Figure 5 marinedrugs-17-00380-f005:**
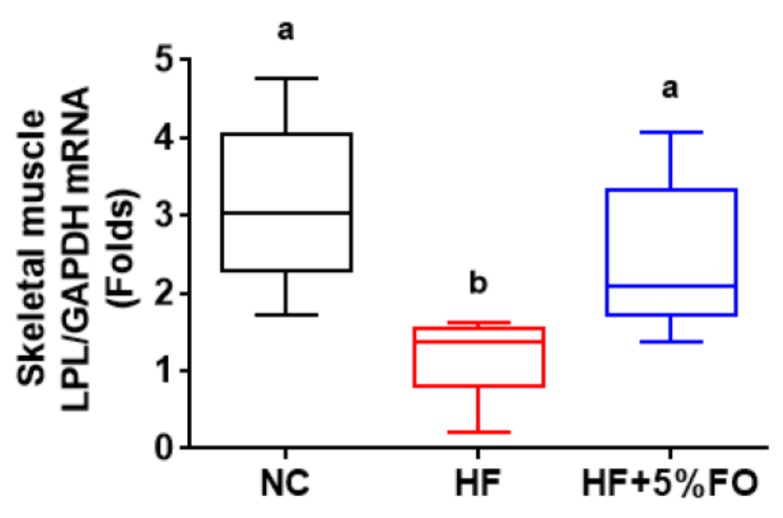
Effects of fish oil on the mRNA expressions of lipoprotein lipase (LPL) in soleus muscles of high-fat diet-fed rats. Rats fed with different experimental diets [standard diet (NC), high-fat diet (HF) in the presence or absence of 5% fish oil (FO)] for eight weeks. Gene expressions of LPL in the soleus muscles were measured by quantitative reverse transcription polymerase chain reaction. All data are expressed as mean ± S.D. for each group (*n* = 6). Values with different letters were significantly different, *P* < 0.05 (ANOVA with post-hoc Tukey’s test).

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
