# Peer review of "Omega-3 Fatty Acids-Enriched Fish Oil Activates AMPK/PGC-1α Signaling and Prevents Obesity-Related Skeletal Muscle Wasting"

_marinedrugs, 2019, doi:10.3390/md17060380_

Reviewer 1 Report

General comments:

In the manuscript entitled “Omega-3 fatty acids-enriched fish oil activates AMPK/PGC-1α signaling and prevents obesity- related skeletal muscle wasting” the authors Liu et al studied the effects of supplementation of 5% omega-3 fatty acid fish oil during 8 weeks in rats receiving high-fat diet. The Authors evaluated this effect analyzing in soleus muscles the protein expression level of adenosine monophosphate (AMP)-activated protein kinase (AMPK) phosphorylation, peroxisome-proliferator-activated receptor-γ (PPARγ) coactivator 1α (PGC-1α), and PPARγ protein expression and lipoprotein lipase (LPL) mRNA expression. According to the data analyses of protein catabolism and protein anabolism pathways, the Authors suggested that supplementation of 5% FO prevents the HF diet-induced muscle wasting.

This original manuscript is a continuity of a previous paper already published where the aim is to investigated the effect and mechanism of fish oil on skeletal muscle metabolism and muscle wasting in HF diet-fed rats. The topic is relevant and interesting, with great value for basic and translational research. The manuscript is well written and clear, with a good logic of presenting and discussing the data.  However, there are some overstatements in the manuscript as well lacking relevant information and discussion that the Authors should address. Moreover, the Authors should consider to change the bars graph to box-plot and present evidences that the data analyses could be done using parametric tests.

It has been extensively shown in humans and animal models that long term HF diet results in an excessive accumulation of adipose tissue in skeletal muscle. In the skeletal muscle system, this leads to muscle atrophy via activation of proteins of the atrophy pathway (TNFα-TNF-R-NFκB-MuRF-1); as consequence, not only body weight increases but also the ubiquitin proteasome system, autophagy, and apoptosis pathways are activated, resulting in a reduction in muscle diameter, specific force and thus percentage of muscle strength. These results are supported by the inverse relationship between fiber size and loss in force generation capacity in in vitro muscle fibers in obese older mice and rats.

Suggested evidences

High-fat diet suppresses the positive effect of creatine supplementation on skeletal muscle function by reducing protein expression of IGF-PI3K-AKT-mTOR pathway. Ferretti R, Moura EG, Dos Santos VC, Caldeira EJ, Conte M, Matsumura CY, Pertille A, Mosqueira M. PLoS One. 2018 Oct 4;13(10):e0199728. doi: 10.1371/journal.pone.0199728. eCollection 2018.

Blaauw B, Schiaffino S, Reggiani C. Mechanisms modulating skeletal muscle phenotype. Compr Physiol.

2013; 3(4):1645±87. https://doi.org/10.1002/cphy.c130009 PMID: 24265241.

Roy B, Curtis ME, Fears LS, Nahashon SN, Fentress HM. Molecular Mechanisms of Obesity-Induced

Osteoporosis and Muscle Atrophy. Front Physiol. 2016; 7:439. https://doi.org/10.3389/fphys.2016.

00439 PMID: 27746742; PubMed Central PMCID: PMCPMC5040721.

Abrigo J, Rivera JC, Aravena J, Cabrera D, Simon F, Ezquer F, et al. High Fat Diet-Induced Skeletal

Muscle Wasting Is Decreased by Mesenchymal Stem Cells Administration: Implications on Oxidative

Stress, Ubiquitin Proteasome Pathway Activation, and Myonuclear Apoptosis. Oxid Med Cell Longev.

2016; 2016:9047821. https://doi.org/10.1155/2016/9047821 PMID: 27579157; PubMed Central

PMCID: PMCPMC4992759.

Ma J, Hwang SJ, McMahon GM, Curhan GC, McLean RR, Murabito JM, et al. Mid-adulthood cardiometabolic

risk factor profiles of sarcopenic obesity. Obesity (Silver Spring). 2016; 24(2):526±34. https://doi.

org/10.1002/oby.21356 PMID: 26813531.

Sala D, Ivanova S, Plana N, Ribas V, Duran J, Bach D, et al. Autophagy-regulating TP53INP2 mediates

muscle wasting and is repressed in diabetes. J Clin Invest. 2014; 124(5):1914±27. https://doi.org/10.

1172/JCI72327 PMID: 24713655; PubMed Central PMCID: PMCPMC4001546.

Shortreed KE, Krause MP, Huang JH, Dhanani D, Moradi J, Ceddia RB, et al. Muscle-specific adaptations,

impaired oxidative capacity and maintenance of contractile function characterize diet-induced

obese mouse skeletal muscle. PLoS One. 2009; 4(10):e7293. https://doi.org/10.1371/journal.pone.

0007293 PMID: 19806198; PubMed Central PMCID: PMCPMC2752162.

Sishi B, Loos B, Ellis B, Smith W, du Toit EF, Engelbrecht AM. Diet-induced obesity alters signalling

pathways and induces atrophy and apoptosis in skeletal muscle in a prediabetic rat model. Exp Physiol.

2011; 96(2):179±93. https://doi.org/10.1113/expphysiol.2010.054189 PMID: 20952489.

Choi SJ, Files DC, Zhang T, Wang ZM, Messi ML, Gregory H, et al. Intramyocellular Lipid and Impaired

Myofiber Contraction in Normal Weight and Obese Older Adults. J Gerontol A Biol Sci Med Sci. 2016;

71(4):557±64. https://doi.org/10.1093/gerona/glv169 PMID: 26405061; PubMed Central PMCID:

PMCPMC5014190.

Eshima H, Tamura Y, Kakehi S, Kurebayashi N, Murayama T, Nakamura K, et al. Long-term, but not

short-term high-fat diet induces fiber composition changes and impaired contractile force in mouse fasttwitch

skeletal muscle. Physiol Rep. 2017; 5(7). https://doi.org/10.14814/phy2.13250 PMID: 28408640;

PubMed Central PMCID: PMCPMC5392533.

The authors focus the protein analysis in few proteins of the degradation signaling pathway, but did not analyze the effect of FO on the protein the synthesis signaling pathway IGF1-IRS1-PI3K-AKT-mTOR pathway. As an example, in a meta-analysis, it has been shown that CrM supplementation during resistance training increased lean tissue mass by ca. 1.4kg resulting in a significant increase in force in comparison to placebo. The mechanism that creatine increases muscle mass and force is increasing the expression of insulin-like growth factor-1 (IGF-1), which would activate the key elements of protein synthesis of the IGF1-IRS1-PI3K-AKT-mTOR pathway. The resultant increase of IGF-1 via creatine is also observable in the significantly increased expression of several myogenic regulatory factors, such as Myo-D, Myf-5 and MRF-4, which are responsible for synchronized triggering of satellite cell activation, proliferation and differentiation. The Authors therefore should discuss how the FO would improve or revert the negative effect caused by the HF diet.

Suggested references

High-fat diet suppresses the positive effect of creatine supplementation on skeletal muscle function by reducing protein expression of IGF-PI3K-AKT-mTOR pathway. Ferretti R, Moura EG, Dos Santos VC, Caldeira EJ, Conte M, Matsumura CY, Pertille A, Mosqueira M. PLoS One. 2018 Oct 4;13(10):e0199728. doi: 10.1371/journal.pone.0199728. eCollection 2018.

Chilibeck PD, Kaviani M, Candow DG, Zello GA. Effect of creatine supplementation during resistance

training on lean tissue mass and muscular strength in older adults: a meta-analysis. Open Access J

Sports Med. 2017; 8:213±26. https://doi.org/10.2147/OAJSM.S123529 PMID: 29138605; PubMed

Central PMCID: PMCPMC5679696.

Burke DG, Candow DG, Chilibeck PD, MacNeil LG, Roy BD, Tarnopolsky MA, et al. Effect of creatine

supplementation and resistance-exercise training on muscle insulin-like growth factor in young adults.

Int J Sport Nutr Exerc Metab. 2008; 18(4):389±98. PMID: 18708688.

Deldicque L, Theisen D, Bertrand L, Hespel P, Hue L, Francaux M. Creatine enhances differentiation of

myogenic C2C12 cells by activating both p38 and Akt/PKB pathways. Am J Physiol Cell Physiol. 2007;

293(4):C1263±71. https://doi.org/10.1152/ajpcell.00162.2007 PMID: 17652429.

Fujita S, Dreyer HC, Drummond MJ, Glynn EL, Cadenas JG, Yoshizawa F, et al. Nutrient signalling in

the regulation of human muscle protein synthesis. J Physiol. 2007; 582(Pt 2):813±23. https://doi.org/10.

1113/jphysiol.2007.134593 PMID: 17478528; PubMed Central PMCID: PMCPMC2075348.

Zanou N, Gailly P. Skeletal muscle hypertrophy and regeneration: interplay between the myogenic regulatory factors (MRFs) and insulin-like growth factors (IGFs) pathways. Cell Mol Life Sci. 2013; 70

(21):4117±30. https://doi.org/10.1007/s00018-013-1330-4 PMID: 23552962.

Louis M, Van Beneden R, Dehoux M, Thissen JP, Francaux M. Creatine increases IGF-I and myogenic

regulatory factor mRNA in C(2)C(12) cells. FEBS Lett. 2004; 557(1±3):243±7. PMID: 14741375.

Deldicque L, Louis M, Theisen D, Nielens H, Dehoux M, Thissen JP, et al. Increased IGF mRNA in

human skeletal muscle after creatine supplementation. Med Sci Sports Exerc. 2005; 37(5):731±6.

PMID: 15870625.

Detailed comments:

1)                The Authors are overstating the aim of the manuscript saying “we further investigated the effect and mechanism of fish oil on skeletal muscle metabolism and muscle wasting in HF diet-fed rats.” Based on the results presented here, there is no mechanism that allow to explain the effect of FO observed. The Authors should write the aim accordingly to what it was presented in the manuscript.

2)                Regarding Figure 1, the Authors should, whenever it is possible, show the progression of the body weight in the NC, HF and HF+5%FO diets.

3)                Regarding Figure 1, there's no quantification of the CSA data. The Authors should use minimal Feret's fiber diameter.  It has been previously report that higher minimal Feret's diameter on gastrocnemius muscle's fiber after HF diet, with a shift to the right on the minimal Feret's diameter showing a change from slow to fast myofibers type. The Authors should not only analyze the CSA using the minimal Feret’s diameter method, but also discuss how the FO would modify the CSA of a slow muscle.

4)      Regarding Figure 1, the Authors should 1) change the data presentation from bars to box plot. 2) show the raw data of body weight. 3) show the normalized data of soleus/body weight and 4) analyzed the CSA according to Ferretti 2018.

Regarding Figure 2, The Authors should describe which MHC was evaluated (slow or fast) and which antibody was used.

5)      From line 97, the Authors must clarify why the phospho-AMPK was measured.

6)      From line 98, the Authors must clarify 1) why mRNA of LPL was measured and not its protein level, 2) what's LPL role in the HF diet effect, 3) normalize LP levels with a proper housekeeping gene and not the levels of HF.

7)      From line 138, the Authors must show the purpose and the data with the collected blood.

8)      From line 149, the Authors must provide 1) composition of blocking solution, 2) dilution of the antibodies used 3) specify the secondary antibody and its/ their dilution.

9)      From line 175, the Authors must clarify 1) whether it was used a parametric or a non-parametric test, 2) whether the primary criterion to use parametric test were achieved; otherwise, the Authors should re-analyze the data using non-parametric statistical test.

10)   From line 268, The graphic abstract is not clear. The Authors should simplify the figure showing the effect of HF diet in the skeletal muscle and then where the fish oil would inhibit the negative effect of HF diet. 

11)   The Authors should consider to discuss the following list of papers.

 J Nutr Biochem. 2018 May;55:76-88. doi: 10.1016/j.jnutbio.2017.11.012. Epub 2017 Dec 11. Attenuation of obesity and insulin resistance by fish oil supplementation is associated with improved skeletal muscle mitochondrial function in mice fed a high-fat diet. Martins AR1, Crisma AR1, Masi LN2, Amaral CL3, Marzuca-Nassr GN4, Bomfim LHM5, Teodoro BG6, Queiroz AL6, Serdan TDA2, Torres RP7, Mancini-Filho J7, Rodrigues AC1, Alba-Loureiro TC1, Pithon-Curi TC2, Gorjao R2, Silveira LR5, Curi R8, Newsholme P9, Hirabara SM10.

Front Nutr. 2018 Mar 12;5:15. doi: 10.3389/fnut.2018.00015. eCollection 2018. Effect of  Eicosapentaenoic Acid and Docosahexaenoic Acid on Myogenesis and Mitochondrial Biosynthesis during Murine Skeletal Muscle Cell Differentiation. Hsueh TY1, Baum JI2, Huang Y1.

PLoS One. 2016 Feb 22;11(2):e0149033. doi: 10.1371/journal.pone.0149033. eCollection 2016. Polyunsaturated Fatty Acids Attenuate Diet Induced Obesity and Insulin Resistance, Modulating Mitochondrial Respiratory Uncoupling in Rat Skeletal Muscle. Cavaliere G1, Trinchese G1, Bergamo P2, De Filippo C1, Mattace Raso G3, Gifuni G1, Putti R1, Moni BH1, Canani RB4, Meli R3, Mollica MP1.

Nutrients. 2016 Sep 3;8(9). pii: E543. doi: 10.3390/nu8090543. Eicosapentaenoic and Docosahexaenoic Acid-Enriched High Fat Diet Delays Skeletal Muscle Degradation in Mice. Soni NK1, Ross AB2, Scheers N3, Savolainen OI4, Nookaew I5,6, Gabrielsson BG7, Sandberg AS8.

Biogerontology. 2017 Feb;18(1):109-129. doi: 10.1007/s10522-016-9667-3. Epub 2016 Nov 18. Omega-3 fatty acid EPA improves regenerative capacity of mouse skeletal muscle cells exposed to saturated fat and inflammation. Saini A1, Sharples AP2, Al-Shanti N3, Stewart CE4.

Exp Physiol. 2017 Nov 1;102(11):1500-1512. doi: 10.1113/EP086317. Epub 2017 Sep 30. Effect of conjugated linoleic acids and omega-3 fatty acids with or without resistance training on muscle mass in high-fat diet-fed middle-aged mice. Lee SR1,2, Khamoui AV2,3,4, Jo E2,3,5, Zourdos MC2,4, Panton LB2,6, Ormsbee MJ2,6, Kim JS2,3,6.

J Exerc Nutrition Biochem. 2017 Sep 30;21(3):11-18. doi: 10.20463/jenb.2017.0028. Effects of conjugated linoleic acid/n-3 and resistance training on muscle quality and expression of atrophy-related ubiquitin ligases in middle-aged mice with high-fat dietinduced obesity. Oh SL1, Lee SR2, Kim JS3.

PLoS One. 2013 Nov 7;8(11):e78874. doi: 10.1371/journal.pone.0078874. eCollection 2013. Effects of dietary eicosapentaenoic acid (EPA) supplementation in high-fat fed mice on lipid metabolism and apelin/APJ system in skeletal muscle. Bertrand C1, Pignalosa A, Wanecq E, Rancoule C, Batut A, Deleruyelle S, Lionetti L, Valet P, Castan-Laurell I.

Author Response

Reviewer 1

1. General comments:

(1). In the manuscript entitled “Omega-3 fatty acids-enriched fish oil activates AMPK/PGC-1α signaling and prevents obesity- related skeletal muscle wasting” the authors Liu et al studied the effects of supplementation of 5% omega-3 fatty acid fish oil during 8 weeks in rats receiving high-fat diet. The Authors evaluated this effect analyzing in soleus muscles the protein expression level of adenosine monophosphate (AMP)-activated protein kinase (AMPK) phosphorylation, peroxisome-proliferator-activated receptor-γ (PPARγ) coactivator 1α (PGC-1α), and PPARγ protein expression and lipoprotein lipase (LPL) mRNA expression. According to the data analyses of protein catabolism and protein anabolism pathways, the Authors suggested that supplementation of 5% FO prevents the HF diet-induced muscle wasting. This original manuscript is a continuity of a previous paper already published where the aim is to investigate the effect and mechanism of fish oil on skeletal muscle metabolism and muscle wasting in HF diet-fed rats. The topic is relevant and interesting, with great value for basic and translational research. The manuscript is well written and clear, with a good logic of presenting and discussing the data.  However, there are some overstatements in the manuscript as well lacking relevant information and discussion that the Authors should address. Moreover, the Authors should consider to change the bars graph to box-plot and present evidences that the data analyses could be done using parametric tests.

Response: We appreciate the reviewer's comment. We have corrected the overstatements and addressed relevant discussion in this revised manuscript according to the suggestion of reviewer. Moreover, we have replaced the bars graph to box-plot in all Figures of this revised manuscript according to the suggestion of reviewer.

(2). It has been extensively shown in humans and animal models that long term HF diet results in an excessive accumulation of adipose tissue in skeletal muscle. In the skeletal muscle system, this leads to muscle atrophy via activation of proteins of the atrophy pathway (TNFα-TNF-R-NFκB-MuRF-1); as consequence, not only body weight increases but also the ubiquitin proteasome system, autophagy, and apoptosis pathways are activated, resulting in a reduction in muscle diameter, specific force and thus percentage of muscle strength. These results are supported by the inverse relationship between fiber size and loss in force generation capacity in in vitro muscle fibers in obese older mice and rats.

Suggested evidences:

High-fat diet suppresses the positive effect of creatine supplementation on skeletal muscle function by reducing protein expression of IGF-PI3K-AKT-mTOR pathway. Ferretti R, Moura EG, Dos Santos VC, Caldeira EJ, Conte M, Matsumura CY, Pertille A, Mosqueira M. PLoS One. 2018 Oct 4;13(10):e0199728. doi: 10.1371/journal.pone.0199728. eCollection 2018.

Blaauw B, Schiaffino S, Reggiani C. Mechanisms modulating skeletal muscle phenotype. Compr Physiol.2013; 3(4):1645±87. https://doi.org/10.1002/cphy.c130009 PMID: 24265241.

 Roy B, Curtis ME, Fears LS, Nahashon SN, Fentress HM. Molecular Mechanisms of Obesity-Induced Osteoporosis and Muscle Atrophy. Front Physiol. 2016; 7:439. https://doi.org/10.3389/fphys.2016.

00439 PMID: 27746742; PubMed Central PMCID: PMCPMC5040721.

 Abrigo J, Rivera JC, Aravena J, Cabrera D, Simon F, Ezquer F, et al. High Fat Diet-Induced Skeletal Muscle Wasting Is Decreased by Mesenchymal Stem Cells Administration: Implications on Oxidative Stress, Ubiquitin Proteasome Pathway Activation, and Myonuclear Apoptosis. Oxid Med Cell Longev.2016; 2016:9047821. https://doi.org/10.1155/2016/9047821 PMID: 27579157; PubMed Central

PMCID: PMCPMC4992759.

 Ma J, Hwang SJ, McMahon GM, Curhan GC, McLean RR, Murabito JM, et al. Mid-adulthood cardiometabolic risk factor profiles of sarcopenic obesity. Obesity (Silver Spring). 2016; 24(2):526±34. https://doi.org/10.1002/oby.21356 PMID: 26813531.

 Sala D, Ivanova S, Plana N, Ribas V, Duran J, Bach D, et al. Autophagy-regulating TP53INP2 mediates muscle wasting and is repressed in diabetes. J Clin Invest. 2014; 124(5):1914±27. https://doi.org/10.1172/JCI72327 PMID: 24713655; PubMed Central PMCID: PMCPMC4001546.

 Shortreed KE, Krause MP, Huang JH, Dhanani D, Moradi J, Ceddia RB, et al. Muscle-specific adaptations, impaired oxidative capacity and maintenance of contractile function characterize diet-induced obese mouse skeletal muscle. PLoS One. 2009; 4(10):e7293. https://doi.org/10.1371/journal.pone.0007293 PMID: 19806198; PubMed Central PMCID: PMCPMC2752162.

 Sishi B, Loos B, Ellis B, Smith W, du Toit EF, Engelbrecht AM. Diet-induced obesity alters signaling pathways and induces atrophy and apoptosis in skeletal muscle in a prediabetic rat model. Exp Physiol. 2011; 96(2):179±93. https://doi.org/10.1113/expphysiol.2010.054189 PMID: 20952489.

 Choi SJ, Files DC, Zhang T, Wang ZM, Messi ML, Gregory H, et al. Intramyocellular Lipid and Impaired Myofiber Contraction in Normal Weight and Obese Older Adults. J Gerontol A Biol Sci Med Sci. 2016; 71(4):557±64. https://doi.org/10.1093/gerona/glv169 PMID: 26405061; PubMed Central PMCID:PMCPMC5014190.

 Eshima H, Tamura Y, Kakehi S, Kurebayashi N, Murayama T, Nakamura K, et al. Long-term, but not short-term high-fat diet induces fiber composition changes and impaired contractile force in mouse fast twitch skeletal muscle. Physiol Rep. 2017; 5(7). https://doi.org/10.14814/phy2.13250 PMID: 28408640; PubMed Central PMCID: PMCPMC5392533.

Response: We appreciate the reviewer's comment. We have addressed more discussion from these studies in the Results & Discussion section of this revised manuscript according to the suggestion of reviewer.

(3). The authors focus the protein analysis in few proteins of the degradation signaling pathway, but did not analyze the effect of FO on the protein the synthesis signaling pathway IGF1-IRS1-PI3K-AKT-mTOR pathway. As an example, in a meta-analysis, it has been shown that CrM supplementation during resistance training increased lean tissue mass by ca. 1.4kg resulting in a significant increase in force in comparison to placebo. The mechanism that creatine increases muscle mass and force is increasing the expression of insulin-like growth factor-1 (IGF-1), which would activate the key elements of protein synthesis of the IGF1-IRS1-PI3K-AKT-mTOR pathway. The resultant increase of IGF-1 via creatine is also observable in the significantly increased expression of several myogenic regulatory factors, such as Myo-D, Myf-5 and MRF-4, which are responsible for synchronized triggering of satellite cell activation, proliferation and differentiation. The Authors therefore should discuss how the FO would improve or revert the negative effect caused by the HF diet.

Suggested references:

High-fat diet suppresses the positive effect of creatine supplementation on skeletal muscle function by reducing protein expression of IGF-PI3K-AKT-mTOR pathway. Ferretti R, Moura EG, Dos Santos VC, Caldeira EJ, Conte M, Matsumura CY, Pertille A, Mosqueira M. PLoS One. 2018 Oct 4;13(10):e0199728. doi: 10.1371/journal.pone.0199728. eCollection 2018.

 Chilibeck PD, Kaviani M, Candow DG, Zello GA. Effect of creatine supplementation during resistance training on lean tissue mass and muscular strength in older adults: a meta-analysis. Open Access J Sports Med. 2017; 8:213±26. https://doi.org/10.2147/OAJSM.S123529 PMID: 29138605; PubMed Central PMCID: PMCPMC5679696.

 Burke DG, Candow DG, Chilibeck PD, MacNeil LG, Roy BD, Tarnopolsky MA, et al. Effect of creatine supplementation and resistance-exercise training on muscle insulin-like growth factor in young adults. Int J Sport Nutr Exerc Metab. 2008; 18(4):389±98. PMID: 18708688.

 Deldicque L, Theisen D, Bertrand L, Hespel P, Hue L, Francaux M. Creatine enhances differentiation of myogenic C2C12 cells by activating both p38 and Akt/PKB pathways. Am J Physiol Cell Physiol. 2007; 293(4):C1263±71. https://doi.org/10.1152/ajpcell.00162.2007 PMID: 17652429.

 Fujita S, Dreyer HC, Drummond MJ, Glynn EL, Cadenas JG, Yoshizawa F, et al. Nutrient signalling in the regulation of human muscle protein synthesis. J Physiol. 2007; 582(Pt 2):813±23. https://doi.org/10.1113/jphysiol.2007.134593 PMID: 17478528; PubMed Central PMCID: PMCPMC2075348.

 Zanou N, Gailly P. Skeletal muscle hypertrophy and regeneration: interplay between the myogenic regulatory factors (MRFs) and insulin-like growth factors (IGFs) pathways. Cell Mol Life Sci. 2013; 70 (21):4117±30. https://doi.org/10.1007/s00018-013-1330-4 PMID: 23552962.

 Louis M, Van Beneden R, Dehoux M, Thissen JP, Francaux M. Creatine increases IGF-I and myogenic regulatory factor mRNA in C(2)C(12) cells. FEBS Lett. 2004; 557(1±3):243±7. PMID: 14741375.

 Deldicque L, Louis M, Theisen D, Nielens H, Dehoux M, Thissen JP, et al. Increased IGF mRNA in human skeletal muscle after creatine supplementation. Med Sci Sports Exerc. 2005; 37(5):731±6.PMID: 15870625.

Response: We appreciate the reviewer's comment. We have addressed more discussion from these studies in the Results & Discussion section of this revised manuscript and tried to explain how the FO would improve or revert the negative effect caused by the HF diet according to the suggestion of reviewer.

2. Detailed comments:

1)   The Authors are overstating the aim of the manuscript saying “we further investigated the effect and mechanism of fish oil on skeletal muscle metabolism and muscle wasting in HF diet-fed rats.” Based on the results presented here, there is no mechanism that allow to explain the effect of FO observed. The Authors should write the aim accordingly to what it was presented in the manuscript.

Response: We appreciate the reviewer's comment. We have re-described the statement of the aim in this revised manuscript according to the suggestion of reviewer.

2)   Regarding Figure 1, the Authors should, whenever it is possible, show the progression of the body weight in the NC, HF and HF+5%FO diets.

Response: We appreciate the reviewer's comment. Since this manuscript is a continuity of our previous publication [ref. 24], the progression of the body weights of animals is the same as our previous study. We have shown the data for body weight changes as the box-plot in the Figure 1A of this revised manuscript according to the suggestion of reviewer.

3)   Regarding Figure 1, there's no quantification of the CSA data. The Authors should use minimal Feret's fiber diameter.  It has been previously report that higher minimal Feret's diameter on gastrocnemius muscle's fiber after HF diet, with a shift to the right on the minimal Feret's diameter showing a change from slow to fast myofibers type. The Authors should not only analyze the CSA using the minimal Feret’s diameter method, but also discuss how the FO would modify the CSA of a slow muscle.

Response: We appreciate the reviewer's comment. We have addressed the quantification of the CSA in soleus muscle analyzed by the methods described previously (Chiu et al., 2016; Ferretti et al., 2018) and described the analytical methods in the Figure 2B and Materials and Methods sections of this revised manuscript according to the suggestion of reviewer.

4)   Regarding Figure 1, the Authors should 1) change the data presentation from bars to box plot. 2) show the raw data of body weight. 3) show the normalized data of soleus/body weight and 4) analyzed the CSA according to Ferretti 2018.

Response: We appreciate the reviewer's comment.

(1)   We have replaced the bars graph to box-plot in Figure 1 of this revised manuscript according to the suggestion of reviewer.

(2)   Since this manuscript is a continuity of our previous publication, the progression of the body weights of animals is the same as our previous study and is shown as the box-plot in the Figure 1A of this revised manuscript according to the suggestion of reviewer.

(3)   We have addressed the result of the soleus muscle weight over body weight in the Figure 1B of this revised manuscript according to the suggestion of reviewer.

(4)   We have addressed the quantification of the CSA in soleus muscle analyzed by the methods described previously (Chiu et al., 2016; Ferretti et al., 2018) and described the analytical methods in the Figure 2B and Materials and Methods sections of this revised manuscript according to the suggestion of reviewer.

References

Chiu, C.Y.; Yang, R.S.; Sheu, M.L.; Chan, D.C.; Yang, T.H.; Tsai, K.S.; Chiang, C.K.; Liu, S.H. Advanced glycation end-products induce skeletal muscle atrophy and dysfunction in diabetic mice via a RAGE-mediated, AMPK-down-regulated, Akt pathway. J. Pathol. 2016, 238(3), 470-82.

Ferretti, R.; Moura, E.G.; Dos Santos, V.C.; Caldeira, E.J.; Conte, M.; Matsumura, C.Y.; Pertille, A.; Mosqueira, M. High-fat diet suppresses the positive effect of creatine supplementation on skeletal muscle function by reducing protein expression of IGF-PI3K-AKT-mTOR pathway. PLoS One 2018, 13(10), e0199728.

5)   Regarding Figure 2, The Authors should describe which MHC was evaluated (slow or fast) and which antibody was used.

Response: We appreciate the reviewer's comment. We have addressed the description of fast MHC protein expression and the information of anti-MHC antibody (A4.1025 antibody, anti-all MHC, Merck Millipore) in the Materials and Methods sections of this revised manuscript according to the suggestion of reviewer.

6)   From line 97, the Authors must clarify why the phospho-AMPK was measured.

Response: We appreciate the reviewer's comment. According to the study of Atherton et al. (2005), the activation of AMPK/mTOR signaling pathway through phosphorylation is correlated with protein synthesis and muscle hypertrophy. Therefore, in the present study, we would investigate the effects of fish oil against HF diet-induced muscle wasting though activating AMPK signaling by phosphorylation.

Reference

Atherton, P.J.; Babraj, J.; Smith, K.; Singh, J.; Rennie, M.J.; Wackerhage, H. Selective activation of AMPK-PGC-1 alpha or PKB-TSC2-mTOR signaling can explain specific adaptive responses to endurance or resistance training-like electrical muscle stimulation. FASEB J. 2005, 19, 786–788.

7)   From line 98, the Authors must clarify 1) why mRNA of LPL was measured and not its protein level, 2) what's LPL role in the HF diet effect, 3) normalize LP levels with a proper housekeeping gene and not the levels of HF.

Response: We appreciate the reviewer's comment.

1.      We could not detect the protein levels of LPL in soleus muscles of each group by the commercial LPL antibody, which may be due to the affinity of the commercial antibody. Therefore, we measured the mRNA levels of LPL.

2.      According to previous studies, high-fat diets or high-carbohydrates contributing to the development of obesity may suppress fat oxidation through down-regulation of LPL expression in skeletal muscle (Jensen et al., 1997; Roberts et al., 2002). Moreover, Boivin et al. (1994) have suggested that insulin resistance induced by HF diets and high-carbohydrates diets in rats may decrease skeletal muscle LPL activity. Therefore, in the present study, FO could reverse HF diet-inhibited LPL gene expression in soleus muscles (Figure 3B).

References

Jensen, D.R.; Schlaepfer, I.R.; Morin, C.L.; Pennington, D.S.; Marcell, T.; Ammon, S.M.; Gutierrez-Hartmann, A.; Eckel, R.H. Prevention of diet-induced obesity in transgenic mice overexpressing skeletal muscle lipoprotein lipase. Am. J. Physiol. 1997, 273(2 Pt 2), R683-9.

Roberts, C.K.; Barnard, R.J.; Liang, K.H.; Vaziri, N.D. Effect of diet on adipose tissue and skeletal muscle VLDL receptor and LPL: implications for obesity and hyperlipidemia. Atherosclerosis 2002, 161(1), 133-41.

Boivin, A.; Montplaisir, I.; Deshaies, Y. Postprandial modulation of lipoprotein lipase in rats with insulin resistance. Am. J. Physiol. 1994, 267(4 Pt 1), E620-7.

3.      We have replaced the box-plot of LPL/GAPDH gene expression in soleus muscle in the Figure 3B of this revised manuscript according to the suggestion of reviewer.

8)   From line 138, the Authors must show the purpose and the data with the collected blood.

Response: We appreciate the reviewer's comment. We have removed the miswritten statement in the Materials and Methods sections of this revised manuscript according to the suggestion of reviewer.

9)   From line 149, the Authors must provide 1) composition of blocking solution, 2) dilution of the antibodies used 3) specify the secondary antibody and its/ their dilution.

Response: We appreciate the reviewer's comment. We have addressed the composition of blocking solution, and the catalogue numbers, companies and their dilution of primary and secondary antibodies in the Materials and Methods sections of this revised manuscript according to the suggestion of reviewer.

Western blotting was determined as described previously by Chiu et al. [24]. Briefly, tissue lysates were collected by using the homogenized buffer (5 mM HEPES, 320 mM sucrose, 150 mM NaCl, 13 mM Tris-HCl, 13 mM EDTA) with a commercial cocktail of protease and phosphatase inhibitors (78443; 1:100 dilution) (Thermo Fisher Scientific, Waltham, MA, USA). After centrifugation, the supernatants were harvested and the protein concentration were determined by the BCA protein assay kit (23225; Thermo Fisher Scientific). Tissue proteins (50-100 μg) and prestained protein ladders (10 to 180 kDa or 250 kDa) (26616 or 26619; Thermo Fisher Scientific) were loaded in SDS-PAGE gel (8%-12%) and then transferred onto polyvinylidene difluoride membranes (Bio-Rad, Hercules, CA, USA). The membranes were blocking with 5 % non-fat dry milk (Fonterra Brands, Taipei, Taiwan) or 3% BSA (Sigma-Aldrich) in 0.2 % TBS-T buffer for at least 1 h, and then probed with primary antibodies for phosphorylated adenosine monophosphate (AMP)-activated protein kinase α (p-AMPKα) (Thr172)  (2535; 1:1000 dilution) (Cell Signaling Technology, Danvers, MA, USA), AMPKα (5831S; 1:1000 dilution) (Cell Signaling Technology), peroxisome-proliferator activated receptor-γ (PPARγ) coactivator 1α (PGC-1α) (ab54481; 1:1000 dilution) (Abcam, Cambridge, MA, USA), PPARγ (sc-7273; 1:1000 dilution) (Santa Cruz Biotechnology, Santa Cruz, CA, USA), myosin heavy chain (MHC) (A4.1025 antibody, anti-all MHC, Merck Millipore, Burlington, Massachusetts, USA), Forkhead O (FOXO) 3A (ab53287; 1:1000 dilution), phosphorylated FOXO3A (Ser253) (ab154786; 1:1000 dilution) (Abcam), MAFbx/Atrogin-1 (ab74023; 1:1000 dilution) (Abcam), and β-actin (sc-47778; 1:1000 dilution) (Santa Cruz Biotechnology) at 4oC overnight. Anti-rabbit (7074S; 1:5000 dilution) (Cell Signaling Technology) and anti-mouse (7076S; 1:5000 dilution) (Cell Signaling Technology) horseradish peroxidase-conjugated secondary antibodies (anti)were then used to probe the membranes. The bindings were determined by an enhanced chemiluminescence kit (BioRad Laboratories, Redmond, WA, USA) and then exposed to X-ray film (Fujifilm, Tokyo, Japan). The densitometric quantification of bands was determined by Image J 1.51 software (National Institutes of Health, Bethesda, MD, USA).

10)      From line 175, the Authors must clarify 1) whether it was used a parametric or a non-parametric test, 2) whether the primary criterion to use parametric test were achieved; otherwise, the Authors should re-analyze the data using non-parametric statistical test.

Response: We appreciate the reviewer's comment. We have re-analyzed the data using the one-way analysis of variance (ANOVA) followed by the Turkey’s test for multiple comparisons with GraphPad Prism V6.0 software in the Materials and Methods sections of this revised manuscript according to the suggestion of reviewer.

11)    From line 268, the graphic abstract is not clear. The Authors should simplify the figure showing the effect of HF diet in the skeletal muscle and then where the fish oil would inhibit the negative effect of HF diet. 

Response: We appreciate the reviewer's comment. We have replaced into a high-resolution graphic abstract and simplified it in the graphic abstract of this revised manuscript according to the suggestion of reviewer.

12)    The Authors should consider to discuss the following list of papers.

 J Nutr Biochem. 2018 May;55:76-88. doi: 10.1016/j.jnutbio.2017.11.012. Epub 2017 Dec 11. Attenuation of obesity and insulin resistance by fish oil supplementation is associated with improved skeletal muscle mitochondrial function in mice fed a high-fat diet. Martins AR1, Crisma AR1, Masi LN2, Amaral CL3, Marzuca-Nassr GN4, Bomfim LHM5, Teodoro BG6, Queiroz AL6, Serdan TDA2, Torres RP7, Mancini-Filho J7, Rodrigues AC1, Alba-Loureiro TC1, Pithon-Curi TC2, Gorjao R2, Silveira LR5, Curi R8, Newsholme P9, Hirabara SM10.

Front Nutr. 2018 Mar 12;5:15. doi: 10.3389/fnut.2018.00015. eCollection 2018. Effect of  Eicosapentaenoic Acid and Docosahexaenoic Acid on Myogenesis and Mitochondrial Biosynthesis during Murine Skeletal Muscle Cell Differentiation. Hsueh TY1, Baum JI2, Huang Y1.

PLoS One. 2016 Feb 22;11(2):e0149033. doi: 10.1371/journal.pone.0149033. eCollection 2016. Polyunsaturated Fatty Acids Attenuate Diet Induced Obesity and Insulin Resistance, Modulating Mitochondrial Respiratory Uncoupling in Rat Skeletal Muscle. Cavaliere G1, Trinchese G1, Bergamo P2, De Filippo C1, Mattace Raso G3, Gifuni G1, Putti R1, Moni BH1, Canani RB4, Meli R3, Mollica MP1.

Nutrients. 2016 Sep 3;8(9). pii: E543. doi: 10.3390/nu8090543. Eicosapentaenoic and Docosahexaenoic Acid-Enriched High Fat Diet Delays Skeletal Muscle Degradation in Mice. Soni NK1, Ross AB2, Scheers N3, Savolainen OI4, Nookaew I5,6, Gabrielsson BG7, Sandberg AS8.

Biogerontology. 2017 Feb;18(1):109-129. doi: 10.1007/s10522-016-9667-3. Epub 2016 Nov 18. Omega-3 fatty acid EPA improves regenerative capacity of mouse skeletal muscle cells exposed to saturated fat and inflammation. Saini A1, Sharples AP2, Al-Shanti N3, Stewart CE4.

Exp Physiol. 2017 Nov 1;102(11):1500-1512. doi: 10.1113/EP086317. Epub 2017 Sep 30. Effect of conjugated linoleic acids and omega-3 fatty acids with or without resistance training on muscle mass in high-fat diet-fed middle-aged mice. Lee SR1,2, Khamoui AV2,3,4, Jo E2,3,5, Zourdos MC2,4, Panton LB2,6, Ormsbee MJ2,6, Kim JS2,3,6.

J Exerc Nutrition Biochem. 2017 Sep 30;21(3):11-18. doi: 10.20463/jenb.2017.0028. Effects of conjugated linoleic acid/n-3 and resistance training on muscle quality and expression of atrophy-related ubiquitin ligases in middle-aged mice with high-fat dietinduced obesity. Oh SL1, Lee SR2, Kim JS3.

PLoS One. 2013 Nov 7;8(11):e78874. doi: 10.1371/journal.pone.0078874. eCollection 2013. Effects of dietary eicosapentaenoic acid (EPA) supplementation in high-fat fed mice on lipid metabolism and apelin/APJ system in skeletal muscle. Bertrand C1, Pignalosa A, Wanecq E, Rancoule C, Batut A, Deleruyelle S, Lionetti L, Valet P, Castan-Laurell I.

Response: We appreciate the reviewer's comment. We have discussed these references in the Discussion section of this revised manuscript according to the suggestion of reviewer.

Reviewer 2 Report

This manuscript by Shing-Hwa Liu et al. has a stated aim to “…investigate(d) the beneficial effect and possible molecular mechanism of fish oil in skeletal muscle wasting in the obese rat model.” Therefore, young rats were given a low fat diet (3% lard) a high fat diet (15% lard) or a third of the lard was replaced with fish oil (10% lard + 5% fish oil) for 8 weeks. The main finding (prevent the loss of muscle mass) is interesting, but the current data are over-interpreted and signaling analysis is not sufficient to make confident conclusions.

General:

1. The authors refer to the fish oil treatment in this study as “supplementation.”  It is not. The authors added fish oil but removed lard. Therefore, the actual experimental design is fat “replacement” or “substitution.” These are terms used in some of the original fish oil studies by Storlien et al. A true supplementation study would just add the fish oil to the high fat diet, i.e. 15% lard + 5% fish oil.  Therefore, either change the phrasing to “replacement” or “substitution” throughout the abstract and manuscript to reflect the real experimental design, or add another group of mice with 15% lard + 5% fish oil to allow analysis of supplementation.

2. Immunoblots for FoxO3 and Atrogin-1 are not sufficient to understand “atrophy-related proteins.” Minimally, the authors should also include MuRF-1 (another muscle-specific ubiquitin ligase) and autophagy-related proteins LC3b and p62.

3. It is not sufficient to blot for phosphorylated AMPKalpha and conclude that down-stream targets are phosphorylated/active. Therefore, the author’s must blot for a specific AMPK target – the most common is total ACC / phosphoACC Ser-79 – or use an antibody for the phospho-AMPK substrate motif.

4. The conclusions are over-interpreted. For instance, the authors have not shown that “fish oil exerts a beneficial improvement in imbalance of lipid and muscle metabolisms in obesity.” In fact, there is no data at all about metabolism, only very limited signaling data. Similarly, the authors have not “demonstrated that AMPK/PGC-1alpha suppressed FoxO3/atrophy-specific gene transcription.” The authors have only shown a very modest change in the phosphorylation status of AMPK, no cause-and-effect. The conclusions must be greatly tempered and limited to the data presented.

Specific:

4. The term Atrogin-1 has been misspelled as Atrogen-1 in the abstract, line 89 and figure 2. Please correct.

5. Figure 2. Immunoblotting for myosin heavy chain quantity is only appropriate and accurate if skeletal muscle is homogenized in a specialized (usually high salt) buffer. See, for example, Cosper, P. F., & Leinwand, L. A. (2012). Myosin heavy chain is not selectively decreased in murine cancer cachexia. International Journal of Cancer. Journal International Du Cancer, 130(11), 2722–2727. http://doi.org/10.1002/ijc.26298.  Therefore, either remove this MHC data or repeat the analysis with myosin proteins extracted in a high salt buffer.

6. Figure 3. Virtually all proteins have multiple phosphorylation sites. State in this figure and the text which particular phosphorylation site was probed for.

7. Line 151. How were the tissues homogenized? In what buffer?

8. Line 151. Was a molecular mass marker run on the gels/transferred to the membranes? If so, state which one. Also, give the calculate mass next the the representative bands of all the proteins in Figures 2 and 3. This helps the reader’s confidence in protein identification.  

9. line 154-157. Give the catalog number for the primary antibodies. These companies have multiple antibodies for each protein.

10. Line 169. Explain how GAPDH was validated as an internal control.

11. Line 177. Was the student’s t-test used as a post-hoc test? If so, this is inappropriate because it does not account for repeated comparisons.  A different post-hoc test (such as Tukey's) must be used.

Author Response

This manuscript by Shing-Hwa Liu et al. has a stated aim to “…investigate(d) the beneficial effect and possible molecular mechanism of fish oil in skeletal muscle wasting in the obese rat model.” Therefore, young rats were given a low fat diet (3% lard) a high fat diet (15% lard) or a third of the lard was replaced with fish oil (10% lard + 5% fish oil) for 8 weeks. The main finding (prevent the loss of muscle mass) is interesting, but the current data are over-interpreted and signaling analysis is not sufficient to make confident conclusions.

1. General:

(1). The authors refer to the fish oil treatment in this study as “supplementation.”  It is not. The authors added fish oil but removed lard. Therefore, the actual experimental design is fat “replacement” or “substitution.” These are terms used in some of the original fish oil studies by Storlien et al. A true supplementation study would just add the fish oil to the high fat diet, i.e. 15% lard + 5% fish oil.  Therefore, either change the phrasing to “replacement” or “substitution” throughout the abstract and manuscript to reflect the real experimental design, or add another group of mice with 15% lard + 5% fish oil to allow analysis of supplementation.

Response: We appreciate the reviewer's comment. We have replaced the term “supplementation” into “substitution” throughout this revised manuscript according to the suggestion of reviewer.

(2). Immunoblots for FoxO3 and Atrogin-1 are not sufficient to understand “atrophy-related proteins.” Minimally, the authors should also include MuRF-1 (another muscle-specific ubiquitin ligase) and autophagy-related proteins LC3b and p62.

Response: We appreciate the reviewer's comment. According to the limit of revision period, we have no enough time to execute more experiments about protein expressions of MuRF-1 and autophagy-related proteins LC3b and p62 in this revised manuscript. Nevertheless, we added a description for limitation of our study that other atrophy-related proteins MuRF-1 (another muscle-specific ubiquitin ligase) and autophagy-related proteins LC3b and p62 can be included.

(3). It is not sufficient to blot for phosphorylated AMPKalpha and conclude that down-stream targets are phosphorylated/active. Therefore, the author’s must blot for a specific AMPK target – the most common is total ACC / phosphoACC Ser-79 – or use an antibody for the phospho-AMPK substrate motif.

Response: We appreciate the reviewer's comment. According to the limit of revision period, we have no enough time to execute more experiments about down-stream genes or proteins of AMPKα to determine the direct cause and effects in this revised manuscript. Nevertheless, we added a description for limitation of our study that down-stream genes or proteins of AMPKα can be included.

(4). The conclusions are over-interpreted. For instance, the authors have not shown that “fish oil exerts a beneficial improvement in imbalance of lipid and muscle metabolisms in obesity.” In fact, there is no data at all about metabolism, only very limited signaling data. Similarly, the authors have not “demonstrated that AMPK/PGC-1alpha suppressed FoxO3/atrophy-specific gene transcription.” The authors have only shown a very modest change in the phosphorylation status of AMPK, no cause-and-effect. The conclusions must be greatly tempered and limited to the data presented.

Response: We appreciate the reviewer's comment. We have modified the description of conclusion and added the descriptions for limitations of our study in conclusion section in this revised manuscript according to the suggestion of reviewer.

Several limitations exist in the current study that other atrophy-related proteins MuRF-1 (another muscle-specific ubiquitin ligase) and autophagy-related proteins LC3b and p62 and down-stream genes or proteins of AMPKα are not included. Those may weaken the evidence of the conclusion.

2. Specific:

(1). The term Atrogin-1 has been misspelled as Atrogen-1 in the abstract, line 89 and figure 2. Please correct.

Response: We appreciate the reviewer's comment. We have corrected the typo in the abstract of this revised manuscript according to the suggestion of reviewer.

(2). Figure 2. Immunoblotting for myosin heavy chain quantity is only appropriate and accurate if skeletal muscle is homogenized in a specialized (usually high salt) buffer. See, for example, Cosper, P. F., & Leinwand, L. A. (2012). Myosin heavy chain is not selectively decreased in murine cancer cachexia. International Journal of Cancer. Journal International Du Cancer, 130(11), 2722–2727. http://doi.org/10.1002/ijc.26298.  Therefore, either remove this MHC data or repeat the analysis with myosin proteins extracted in a high salt buffer.

Response: We appreciate the reviewer's comment. We have addressed the procedure of tissue homogenization and the constitution of tissue homogenized buffer (a high-salt buffer) in the Materials and Methods section of this revised manuscript according to the suggestion of reviewer.

Briefly, tissue lysates were collected by using the homogenized buffer (5 mM HEPES, 320 mM sucrose, 150 mM NaCl, 13 mM Tris-HCl, 13 mM EDTA) with a commercial cocktail of protease and phosphatase inhibitors (78443; 1:100 dilution) (Thermo Fisher Scientific, Waltham, MA, USA). After centrifugation, the supernatants were harvested and the protein concentration were determined by the BCA protein assay kit (23225; Thermo Fisher Scientific).

(3). Figure 3. Virtually all proteins have multiple phosphorylation sites. State in this figure and the text which particular phosphorylation site was probed for.

Response: We appreciate the reviewer's comment. We have addressed the phosphorylation site of AMPKα (Thr 172) in the Materials and Methods section and Figure 4 of this revised manuscript according to the suggestion of reviewer.

The membranes were blocking with 5 % non-fat dry milk (Fonterra Brands, Taipei, Taiwan) or 3% BSA (Sigma-Aldrich) in 0.2 % TBS-T buffer for at least 1 h, and then probed with primary antibodies for phosphorylated adenosine monophosphate (AMP)-activated protein kinase α (p-AMPKα) (Thr172)  (2535; 1:1000 dilution) (Cell Signaling Technology, Danvers, MA, USA).

(4). Line 151. How were the tissues homogenized? In what buffer?

Response: We appreciate the reviewer's comment. We have addressed the procedure of tissue homogenization in the Materials and Methods section of this revised manuscript according to the suggestion of reviewer.

Briefly, tissue lysates were collected by using the homogenized buffer (5 mM HEPES, 320 mM sucrose, 150 mM NaCl, 13 mM Tris-HCl, 13 mM EDTA) with a commercial cocktail of protease and phosphatase inhibitors (78443; 1:100 dilution) (Thermo Fisher Scientific, Waltham, MA, USA). After centrifugation, the supernatants were harvested and the protein concentration were determined by the BCA protein assay kit (23225; Thermo Fisher Scientific).

(5). Line 151. Was a molecular mass marker run on the gels/transferred to the membranes? If so, state which one. Also, give the calculate mass next the representative bands of all the proteins in Figures 2 and 3. This helps the reader’s confidence in protein identification.

Response: We appreciate the reviewer's comment. We have addressed the calculate mass of marker the representative bands of all the proteins in Figures 3 and 4 of this revised manuscript according to the suggestion of reviewer.

(6). Line 154-157. Give the catalog number for the primary antibodies. These companies have multiple antibodies for each protein.

Response: We appreciate the reviewer's comment. We have addressed the composition of blocking solution, and the catalogue numbers, companies and their dilution of primary and secondary antibodies in the Materials and Methods sections of this revised manuscript according to the suggestion of reviewer.

Western blotting was determined as described previously by Chiu et al. [24]. Briefly, tissue lysates were collected by using the homogenized buffer (5 mM HEPES, 320 mM sucrose, 150 mM NaCl, 13 mM Tris-HCl, 13 mM EDTA) with a commercial cocktail of protease and phosphatase inhibitors (78443; 1:100 dilution) (Thermo Fisher Scientific, Waltham, MA, USA). After centrifugation, the supernatants were harvested and the protein concentration were determined by the BCA protein assay kit (23225; Thermo Fisher Scientific). Tissue proteins (50-100 μg) and prestained protein ladders (10 to 180 kDa or 250 kDa) (26616 or 26619; Thermo Fisher Scientific) were loaded in SDS-PAGE gel (8%-12%) and then transferred onto polyvinylidene difluoride membranes (Bio-Rad, Hercules, CA, USA). The membranes were blocking with 5 % non-fat dry milk (Fonterra Brands, Taipei, Taiwan) or 3% BSA (Sigma-Aldrich) in 0.2 % TBS-T buffer for at least 1 h, and then probed with primary antibodies for phosphorylated adenosine monophosphate (AMP)-activated protein kinase α (p-AMPKα) (Thr172)  (2535; 1:1000 dilution) (Cell Signaling Technology, Danvers, MA, USA), AMPKα (5831S; 1:1000 dilution) (Cell Signaling Technology), peroxisome-proliferator activated receptor-γ (PPARγ) coactivator 1α (PGC-1α) (ab54481; 1:1000 dilution) (Abcam, Cambridge, MA, USA), PPARγ (sc-7273; 1:1000 dilution) (Santa Cruz Biotechnology, Santa Cruz, CA, USA), myosin heavy chain (MHC) (A4.1025 antibody, anti-all MHC, Merck Millipore, Burlington, Massachusetts, USA), Forkhead O (FOXO) 3A (ab53287; 1:1000 dilution), phosphorylated FOXO3A (Ser253) (ab154786; 1:1000 dilution) (Abcam), MAFbx/Atrogin-1 (ab74023; 1:1000 dilution) (Abcam), and β-actin (sc-47778; 1:1000 dilution) (Santa Cruz Biotechnology) at 4oC overnight. Anti-rabbit (7074S; 1:5000 dilution) (Cell Signaling Technology) and anti-mouse (7076S; 1:5000 dilution) (Cell Signaling Technology) horseradish peroxidase-conjugated secondary antibodies (anti)were then used to probe the membranes. The bindings were determined by an enhanced chemiluminescence kit (BioRad Laboratories, Redmond, WA, USA) and then exposed to X-ray film (Fujifilm, Tokyo, Japan). The densitometric quantification of bands was determined by Image J 1.51 software (National Institutes of Health, Bethesda, MD, USA).

(7). Line 169. Explain how GAPDH was validated as an internal control.

Response: We appreciate the reviewer's comment. We have corrected the miswritten term “an internal control” into “as a reference gene” in the quantitative real-time PCR experiment of this revised manuscript according to the suggestion of reviewer.

(8). Line 177. Was the student’s t-test used as a post-hoc test? If so, this is inappropriate because it does not account for repeated comparisons.  A different post-hoc test (such as Tukey's) must be used.

Response: We appreciate the reviewer's comment. We have re-analyzed the data using the one-way analysis of variance (ANOVA) followed by the Turkey’s test for multiple comparisons with GraphPad Prism V6.0 software in the Materials and Methods sections of this revised manuscript according to the suggestion of reviewer.

Reviewer 3 Report

The authors present a well designed study to investigate some of the potential signaling pathways that fish oil may be acting through to preserve muscle during High fat feeding. 

The authors should include the changes in weight gain throughout the study or see the raw weights for the animals instead of a % of control. This would allow the readers to see how obese the animal were at the time of euthanasia. 

What was the rationale for using the soleus muscle? This muscle is highly oxidative and is not representative of all fiber types, however, that is not mentioned anywhere in the manuscript.  

In figure 1c the image for the HF it is difficult to see the outlines of the individual fibers. Also was cross-sectional area determined for the fibers. It would be interesting to see a histogram the showing any shift in fiber size. This may be important as it appears that there is edema or fibrosis in the FHF-FO group based on the image presented. 

Check spelling of Atrogin in line 89. 

Were there changes in the phosporylation of FOXO3a? The alterations in total are interesting however it would be beneficial to show alterations in the activation/localization of Foxo3a. 

If possible a lower exposure for total AMPK alpha should be shown the current image is over-exposed. 

The authors should be mindful of their conclusions about lipid metabolism. Only gene expression of one gene involved in lipid metabolism was measured. Additionally no knockout or over-expression experiments were conducted to show that AMPK/PGC-1 alpha was directly regulating atrogin and FOXO3 so caution should be used in making these statements suggesting there was direct cause and effect. 

There is a one paragraph discussion. A more detailed discussion would be beneficial to the readers explaining how the result relate to the literature and what the results mean. 

Author Response

Reviewer 3

The authors present a well-designed study to investigate some of the potential signaling pathways that fish oil may be acting through to preserve muscle during High fat feeding. 

1.      The authors should include the changes in weight gain throughout the study or see the raw weights for the animals instead of a % of control. This would allow the readers to see how obese the animal were at the time of euthanasia. 

Response: We appreciate the reviewer's comment. Since this manuscript is a continuity of our previous publication [ref. 24], the progression of the body weights of animals is the same as our previous study. We have shown the data for body weight changes as the box-plot in the Figure 1A of this revised manuscript according to the suggestion of reviewer.

2.      What was the rationale for using the soleus muscle? This muscle is highly oxidative and is not representative of all fiber types, however, that is not mentioned anywhere in the manuscript.  

Response: We appreciate the reviewer's comment. The soleus muscle has been shown to be a properly experimental muscle type in the senescent rat model (Deruisseau et al., 2005) or chemical-induced muscle wasting mouse model (Chen et al., 2019). We added a description for soleus muscle and references in the Methods section of this revised manuscript.

References:

1. Deruisseau KC, Kavazis AN, Powers SK. Selective downregulation of ubiquitin conjugation cascade mRNA occurs in the senescent rat soleus muscle. Exp Gerontol. 2005 Jun;40(6):526-31.

2. Chen HJ, Wang CC, Chan DC, Chiu CY, Yang RS, Liu SH. Adverse effects of acrolein, a ubiquitous environmental toxicant, on muscle regeneration and mass. J Cachexia Sarcopenia Muscle. 2019 Feb;10(1):165-176.

3.      In figure 1c the image for the HF it is difficult to see the outlines of the individual fibers. Also was cross-sectional area determined for the fibers. It would be interesting to see a histogram the showing any shift in fiber size. This may be important as it appears that there is edema or fibrosis in the FHF-FO group based on the image presented. 

Response: We appreciate the reviewer's comment. We have addressed the quantification of the CSA in soleus muscle analyzed by the methods described previously (Chiu et al., 2016; Ferretti et al., 2018) and described the analytical methods in the Figure 2B and Materials and Methods sections of this revised manuscript according to the suggestion of reviewer.

References:

Chiu, C.Y.; Yang, R.S.; Sheu, M.L.; Chan, D.C.; Yang, T.H.; Tsai, K.S.; Chiang, C.K.; Liu, S.H. Advanced glycation end-products induce skeletal muscle atrophy and dysfunction in diabetic mice via a RAGE-mediated, AMPK-down-regulated, Akt pathway. J. Pathol. 2016, 238(3), 470-82.

Ferretti, R.; Moura, E.G.; Dos Santos, V.C.; Caldeira, E.J.; Conte, M.; Matsumura, C.Y.; Pertille, A.; Mosqueira, M. High-fat diet suppresses the positive effect of creatine supplementation on skeletal muscle function by reducing protein expression of IGF-PI3K-AKT-mTOR pathway. PLoS One 2018, 13(10), e0199728.

4.      Check spelling of Atrogin in line 89. 

Response: We appreciate the reviewer's comment. We have corrected the typo in the abstract of this revised manuscript according to the suggestion of reviewer.

5.      Were there changes in the phosporylation of FOXO3a? The alterations in total are interesting however it would be beneficial to show alterations in the activation/localization of Foxo3a. 

Response: We appreciate the reviewer's comment. We have addressed the results of phosphorylated FOXO3a and the ratio over total FOXO3a in Figure 3 of this revised manuscript according to the suggestion of reviewer.

6.      If possible a lower exposure for total AMPK alpha should be shown the current image is over-exposed. 

Response: We appreciate the reviewer's comment. We have provided the optimal exposure image of total AMPKα expression in Figure 4 of this revised manuscript according to the suggestion of reviewer.

7.      The authors should be mindful of their conclusions about lipid metabolism. Only gene expression of one gene involved in lipid metabolism was measured. Additionally no knockout or over-expression experiments were conducted to show that AMPK/PGC-1 alpha was directly regulating atrogin and FOXO3 so caution should be used in making these statements suggesting there was direct cause and effect. 

Response: We appreciate the reviewer's comment. We have modified the description of conclusion. The description for lipid metabolism was deleted.

8.      There is a one paragraph discussion. A more detailed discussion would be beneficial to the readers explaining how the result relate to the literature and what the results mean. 

Response: We appreciate the reviewer's comment. We combined Results and Discussion in a section in this manuscript. We have addressed a more detailed discussion in this revised manuscript according to the suggestion of reviewer.

Round  2

Reviewer 2 Report

The changes are acceptable.